# The Role of Amino Acids in Endothelial Biology and Function

**DOI:** 10.3390/cells11081372

**Published:** 2022-04-18

**Authors:** Meng Li, Yanqing Wu, Lei Ye

**Affiliations:** 1Department of Cardiovascular Medicine, The Second Affiliated Hospital of Nanchang University, Nanchang 330006, China; m13657004331@163.com; 2Department of Biomedical Engineering, University of Alabama at Birmingham, Birmingham, AL 35233, USA

**Keywords:** amino acids, endothelium, endothelial function, metabolism, nitric oxide

## Abstract

The vascular endothelium acts as an important component of the vascular system. It is a barrier between the blood and vessel wall. It plays an important role in regulating blood vessel tone, permeability, angiogenesis, and platelet functions. Several studies have shown that amino acids (AA) are key regulators in maintaining vascular homeostasis by modulating endothelial cell (EC) proliferation, migration, survival, and function. This review summarizes the metabolic and signaling pathways of AAs in ECs and discusses the importance of AA homeostasis in the functioning of ECs and vascular homeostasis. It also discusses the challenges in understanding the role of AA in the development of cardiovascular pathophysiology and possible directions for future research.

## 1. Introduction

The vascular endothelium is a single layer of flattened cells distributed in the inner layer of blood vessels. There are approximately 10 billion ECs in an adult, covering an area of more than 4000 m^2^ and accounting for 1.5% of total body weight [1]. The luminal side of the endothelium directly contacts the blood or lymphatic fluid, while the basal side is connected to the subendothelial tissue by substrates (collagen, elastin, fibronectin, etc.) [2]. The unique anatomical location of the vascular endothelium allows it to not only serve as an important barrier separating the blood from tissues and organs but also integrate physical and neurohumoral signals from blood and surrounding tissues and organs to regulate vascular tone, coagulation, substance exchange, immune responses, and angiogenesis, which are important pathophysiological features of various diseases [3,4]. Under normal physiological conditions, ECs are in a long-term quiescent state, and when stimulated by ischemia, hypoxia, or inflammation, ECs exhibit migration, proliferation, and angiogenesis. In recent years, numerous studies have identified key roles of AA metabolism in vascular system lesions. In this article we discuss the most widely studied AAs and their role in the light of endothelial function.

## 2. Amino Acid in Biology and Function of Endothelial Cells

AAs are organic compounds containing basic amino and acidic carboxyl groups. There are 500 different AAs found in nature, only 20 of which are involved in constituting the proteins required for animal nutrition [5]. According to nutritional classification, AAs can be classified as essential AAs (externally ingested, EAA), conditionally essential AAs (self-synthesized and externally ingested), and non-essential AAs (self-synthesized, NEAA). In humans, AAs are involved in the synthesis of tissue proteins and can be converted into fat for energy storage, oxidized into carbon dioxide and water for energy, or transformed into enzymes, hormones, antibodies, and acids to participate in the regulation of the body fluid, which is crucial for metabolism and growth.

### 2.1. Non-Essential Amino Acids

#### 2.1.1. Glycine

Glycine is an AA with the lowest molecular weight and simplest structure. It is extremely hydrophilic. Glycine is considered an NEAA in the human body. However, premature infants and adults on a low-protein diet do not synthesize enough glycine to meet the body’s daily needs and exogenous supplementation is required [6,7]. As a result, it can also be considered a conditional EAA. Glycine is primarily produced endogenously from serine in the liver via serinehydroxymethyltransferase-2 (SHMT) in the mitochondrial matrix [8,9] (Figure 1). Besides being incorporated into proteins, glycine also has roles in neurotransmission inhibition, cell proliferation, and cell and organ protection [10,11,12,13,14]. Glycine transporters (GlyT) and glycine receptors (GlyR) have been detected on ECs, suggesting that glycine may be involved in the regulation of endothelial biology and function [15,16].

Over the past two decades, glycine has been shown to regulate endothelial function through the following pathways: (i) inducing the expression of the anti-apoptotic protein Bcl2 on the mitochondrial membrane and inhibiting EC apoptosis [17], (ii) inhibiting nuclear factor-kappaB (NF-κB) signaling to downregulate the expression of inflammatory factors such as tumor necrosis factor (TNF)-α and interleukin (IL)-1β, (iii) increasing nitric oxide (NO) bioavailability [18,19], and (iv) promoting glutathione synthesis (GSH) to exert anti-oxidative stress, which protects the vascular endothelium [20,21]. Yu et al. investigated the relationship between glycine and endothelial metabolism and found that exogenous glycine induced EC proliferation and survival and inhibited angiogenesis [22]. Glycine can inhibit EC growth and exert anti-angiogenic effects by binding GlyR on ECs to downregulate vascular endothelial growth factor (VEGF) signaling [23,24]. In addition, VEGF also stimulates EC migration, proliferation, and angiogenesis via the GLT1-glycine-mammalian target of rapamycin (mTOR)-voltage-dependent anion channel 1 (VDAC1) axis [25]. These contrasting effects of glycine were thought to be related to the concentration of glycine. To test this hypothesis, Tsuji-Tamura et al. [26], determined the effect of different glycine concentrations in vascular development and found that glycine had a dose-dependent biphasic effect on vascular development: low glycine concentration induced angiogenesis and high concentration exerted an anti-angiogenic effect. Furthermore, the biphasic effect was found to be associated with the phosphatidylinositol 3-kinase (PI3K)/Akt/mTOR signaling pathway inhibition of the PI3K/Akt/mTOR pathway inhibits the angiogenic effects of low glycine concentrations but acts synergistically with high glycine concentrations and promotes anti-angiogenesis [27]. 

#### 2.1.2. Proline

Proline is a cyclic and nonpolar AA. Like glycine, proline is also an important constituent of collagen and has been reported to account for approximately 10% of all AAs in collagen [28]. Proline is a multifunctional AA that plays an important role in the regulation of dehydration stress, redox, cell proliferation, differentiation, and apoptosis [29,30]. Proline is mainly derived from glutamine but intermediate products for proline synthesis can also be obtained through the tricarboxylic acid (TCA) cycle and urea cycle [28]. Glutamate derived from glutamine metabolism and the TCA cycle produces glutamic-γ-semialdehyde catalyzed by pyrrololine-5-carboxylate synthetase, which is spontaneously cyclized to 1-pyrrololine-5 carboxylate (P5C) [28]. P5C can also be produced from arginine catalyzed by arginase and ornithine aminotransferase. Finally, in the presence of nicotinamide adenine dinucleotide phosphate (NADPH), P5C is converted to proline by the action of pyrrololine-5 carboxylate reductase (P5CR, also known as PYCR in humans), a process closely related to the activation of glycolysis [31]. P5C is also an intermediate product of proline metabolism. Proline is broken down to P5C catalyzed by proline dehydrogenase/proline oxidase (PRODH/POX), and P5C is then converted to glutamate by the action of POX [30], and finally enters the TCA cycle (Figure 1). 

Collagen, the most abundant protein in the body, is essential for the normal structure and strength of connective tissues such as skin, cartilage, and blood vessels. Increased activity of arginase I and II in ECs has been shown to lead to increased production of proline [32]. As proline is an important component of collagen and the extracellular matrix, it may play an important role in vascular remodeling [32]. The ability of proline analogs to induce EC migration and angiogenic responses was discovered as early as the 19^th^ century [33]. Since then, further studies have found that proline can upregulate mRNAs of VEGF, VEGFR, NOS2, and NOS3 in the mouse placenta, indicating that proline is involved in the regulation of vascular development [34]. 

#### 2.1.3. Serine

Serine is an important NEAA discovered by the German chemist Emil Cramer in 1865. L-serine has a very high synthesis rate; it can undergo secondary conversion and act as a precursor for the synthesis of many substances; for example, serine is converted to glycine catalyzed by SHMT and to D-serine catalyzed by serine racemase [35,36]. In humans, L-serine can be derived from food, the conversion of glycine, degradation of proteins and phospholipids, and the serine synthesis pathway (SSP); of these, the SSP is the primary source of L-serine. The intermediate product of glycolysis, 3-phosphoglycerate (3-PG), initiates the SSP [36,37]. 3-PG is converted to L-serine catalyzed by phosphoglycerate dehydrogenase (PHGDH), phosphoserine aminotransferase 1, and phosphoserine phosphatase (PSPH) [37] (Figure 1). 

The SSP is an important metabolic pathway for growth and development. Abnormalities in the SSP lead to multiple organs and vascular dysfunction [38]. In neonatal mouse ECs lacking PHGDH (a key enzyme of the SSP pathway), electron transport chain (ETC) enzyme activity is inhibited, mitochondrial dysfunction is observed, and EC proliferation and survival are reduced, leading to abnormal vascular development [38]. This effect is related to the dependence of EC on SSP for heme synthesis to maintain mitochondrial respiration and homeostasis [39]. The treatment of ECs with L-serine followed by oxidative stress leads to increased expression of antioxidant factors such as nuclear factor-erythroid-related 2 factor (NRF2), heme oxygenase-1 (HO-1), and NO, and increased EC viability and survival [40], suggesting the role of serine in mediating oxidative stress response. L-serine can also regulate endoplasmic reticulum activity, downregulate inflammatory responses and oxidative stress, and improve endothelial function by inhibiting homocysteine (HCY) uptake by ECs [41]. In addition, studies have shown that acute intravenous administration of L-serine in mice activates calcium-activated potassium (K_Ca_) channels on ECs that promote vasodilation in small arteries, leading to antihypertensive effects [42,43,44]. Numerous studies have shown the cardioprotective effects of L-serine in animals. Whether exogenous administration of serine in humans can produce the same clinical effect as in animal models is yet to be determined.

#### 2.1.4. Cysteine

Cysteine, a sulfur-containing AA, is involved in the regulation of protein secondary structure and conformation. Cysteine occupies the active site of several enzymes that can be regulated by redox reactions of sulfhydryl residues to protect cells from oxidative stress [45]. In addition to participating in regulating protein structure, cysteine plays a role in the synthesis of redox cofactors, such as glutathione and hydrogen sulfide (H_2_S), which are important in regulating intracellular oxidative stress [45]. 

Excessive production of reactive oxygen species (ROS) is a leading cause of endothelial dysfunction (ED) and the progression of cardiovascular disease in diabetic patients. ROS can induce leukocytes to adhere to the vessel wall, changing the signaling of the vascular endothelium and leading to increased permeability of the vessel wall [46]. Studies have shown that oral cysteine prevents NF-κB activation and reduces the expression of vascular inflammatory cytokines by inhibiting ROS production, improving diabetes-induced vascular inflammation in rat models of diabetes [18,47]. In high glucose-induced EC injury, exogenous treatment with cysteine protected ECs from oxidative stress injury by increasing glutathione production and downregulating the expression of intercellular adhesion molecule-1 (ICAM-1), vascular cell adhesion molecule-1 (VCAM-1), and monocyte-EC adherence [48,49,50]. Cysteine is an important component of dietary protein and plays an important role in regulating the function of blood vessels. In a study assessing the effects of the 20 standard AAs on the growth of human corneal ECs, L-cysteine deficiency induced apoptosis of corneal ECs [51]. Alban et al. found that restriction of the intake of sulfur-containing AAs such as cysteine in mice leads to the activation of the general control non-repressed 2 (GCN2)/activating transcription factor 4 (ATF4) in ECs [52,53], leading to an upregulation of VEGF expression and increased expression of the trans-sulfur pathway enzyme cystathionine-γ-lyase (CSE), leading to increased production of H2S that promotes angiogenesis by inhibiting mitochondrial electron transport and oxidative phosphorylation [52,53]. 

Endogenous H2S is mainly produced by cysteine in the presence of cystathionine-β-synthase (CBS) and CSE. Both H_2_S enzymes are highly expressed in the vascular system such as vascular ECs and vascular smooth muscle cells [54,55,56]. H_2_S has been shown to induce vasodilation, improve vascular permeability, and promote angiogenesis [55,57,58,59,60,61]. Notably, there is a synergistic effect between H_2_S and NO; H_2_S activates vascular NO signaling. Similarity, impairment of eNOS/cGMP signaling led to a significant reduction in the cardiovascular protective effect of H_2_S [62].

#### 2.1.5. Glutamine and Asparagine 

Glutamine is the most abundant free AA in the body, accounting for approximately 50% of total free AAs in the plasma. It is mainly synthesized from glutamate and ammonia (NH3) under the action of glutamine synthetase (GS) [63]. Besides being incorporated into proteins, glutamine catabolism provides a large source of energy and carbon/nitrogen for cells. Glutamine is catabolized into glutamate and NH3 by glutaminase. Glutamate is metabolized to α-ketoglutarate (α-KG) by the action of glutamate dehydrogenase; α-KG is used as a carbon source to supplement the TCA cycle for the synthesis of NEAAs and lipids or to produce adenosine triphosphate (ATP) (oxidation of 1 mol of glutamine yields 1 mol of GTP and 5 mol of ATP) [64]. In addition, glutamate is used in the synthesis of glutathione, and it is also converted to ornithine to produce NO and polyamines (Figure 1).

The glycolytic pathway’s lactate production/glucose consumption ratio is 2, but in human umbilical vein ECs (HUVEC) it is only 0.1, indicating that glycolysis is not the primary source of energy in HUVEC [64]. One study discovered that when glucose was unavailable, EC’s oxidative capacity for glutamine and glutamate increased [65]. Furthermore, of all AAs, glutamine is the most readily consumed by ECs during proliferation [66]. Intracellular ATP levels in ECs were significantly reduced when glutamine was depleted or catabolism was inhibited, and HUVECs proliferation was inhibited and exhibited a senescence-like phenotype [64,67,68], implying that glutamine catabolism may be an important energy source for ECs. It has been reported that glutamine metabolism accounts for 70% of the carbon supply to the TCA cycle in HUVECs, while glucose metabolism accounts for only 20%. Impaired glutamine metabolism leads to a significant decrease in the TCA intermediate α-KG, leading to TCA cycle blockage. TCA cycle blockage in ECs leads to an inability in biomass synthesis, resulting in decreased EC proliferation [67]. Exogenous supplementation of α-KG in HUVEC cultured in glutamine-free medium only partially restored the proliferative capacity of ECs, which may be because unlike glutamine, α-KG, does not provide a nitrogen source for biomass synthesis. Huang et al. [66] showed that glutamine-derived nitrogen is involved in the synthesis of asparagine. Several studies have shown that glutamine promotes the repair of vascular endothelial function by downregulating inflammatory responses, inhibiting oxidative stress, improving mitochondrial function, mobilizing peripheral circulating endothelial progenitor cells, and inducing the expression of heat shock protein 70 [69,70,71,72].

Asparagine is an AA with a structure very similar to that of glutamine. In most species, asparagine or glutamine can be deamidated by the action of asparaginase or glutaminase and be supplied to the TCA cycle [73]. α-KG combined with asparagine treatment has been shown to completely restore ED due to impaired glutamine metabolism [66]. Pavlova et al., [73] also showed that exogenous supplementation of asparagine rescued glutamine-deficient ED proliferation. These findings suggest that asparagine may play an important role in angiogenesis. Consistent with this hypothesis, EC proliferation was impaired when the expression of asparagine synthase (the enzyme that catalyzes asparagine synthesis) was inhibited [66]. Supplementation with asparagine activates the mTOR pathway and inhibits the endoplasmic reticulum stress response to promote EC proliferation. In addition, the metabolic process of glutamine is also involved in the aging of EC [64]. 

NH3 has long been considered a toxic substance as a high NH3 concentration can cause saponification reactions, leading to tissue mucosa damage. Interestingly, NH3 produced during glutamine metabolism has been shown to promote EC survival by inducing HO-1 expression in ECs via activation of the ROS-NRF2 pathway; HO-1 exerts protective effects on ECs by producing carbon monoxide [74].

#### 2.1.6. Arginine

Arginine belongs to the glutamic acid family. It is synthesized using glutamic acid as a substrate catalyzed by a variety of enzymes. Arginine is largely considered a conditionally EAA because it cannot be synthesized in preterm infants and needs to be obtained exogenously [75]. Arginine is metabolized in the body in the following two ways: (i) it is broken down into urea and ornithine by arginase, or (ii) it is broken down into equal molecules of citrulline and NO by NO synthase (NOS) (Figure 1). Urea is excreted as a metabolic end product, while ornithine can enter the urea cycle as an intermediate product or as a precursor for the synthesis of polyamines that are important for regulating cell growth and development. 

The arginine-NO pathway plays an important role in vascular endothelial function. NO is a small signaling molecule and the most potent endogenous vasodilator known. In vascular ECs, NO can diffuse through the guanylate cyclase pathway to vascular smooth muscle cells, and cause vasodilation and an increase in blood flow. Additionally, NO regulates vascular endothelial function by other aspects such as modulating EC activity, proliferation, migration, and angiogenesis [76]. 

The Michaelis–Menten constant (Km) of NOS has been reported to be around 2.9 μM in the plasma of healthy adults [77], but the concentration of arginine ranges from 0.8 to 2 mM, which far exceeds the value of Km = 2.9 μM. This difference suggests that NOS is in a saturated state under normal physiological conditions and arginine supplementation may not lead to a further increase in NO production. However, exogenous arginine supplementation increases NO production even when NOS is supersaturated [78,79,80], a phenomenon known as the arginine paradox. This may be due to the presence of a NOS inhibitor, asymmetric dimethylarginine (ADMA), which binds NOS competitively. Arginine supplementation further increases the concentration of arginine in the body, thereby increasing the rate of arginine binding to NOS. Therefore, based on this phenomenon, arginine supplementation may be a potential therapeutic approach for the treatment of vascular diseases. However, some studies have shown that arginine supplementation does not increase NO production in healthy subjects [81,82]. Moreover, arginine exerts toxic effects by promoting the formation of NO and peroxynitrite (ONOO-) [83]. Therefore, findings in this regard are contradictory; arginine activity may be dependent on its dose and form of supplementation. However, the precise mechanism of action of arginine is not well understood and needs to be studied further. 

### 2.2. Essential Amino Acids

EAAs are those that cannot be synthesized in the body or cannot be synthesized at a rate sufficient to meet the body’s needs and must be supplied via diet. Most EAAs, such as histidine, lysine, methionine, phenylalanine, threonine, and tryptophan, are degraded primarily in the liver, while the degradation of branched-chain AAs (BCAAs), namely, isoleucine, leucine, and valine, occur primarily in the kidney and muscle. These EAAs can produce glucose via gluconeogenesis pathway, provide energy through oxidation, and serve as a source of nitrogen for other molecules.

#### 2.2.1. Tryptophan, Methionine, and Phenylalanine 

Tryptophan is an EAA that is obtained via the diet. Tryptophan absorbed through intestinal digestion enters the systemic circulation through the portal vein [84]. Tryptophan is partly used for protein synthesis and partly degraded. It has been shown that tryptophan can be catabolized via the kynurenine pathway (KP) and the serotonin pathway, with the KP accounting for 95% of tryptophan catabolism [84]. Three important rate-limiting enzymes, namely, indoleamine 2,3-dioxygenase 1 and 2 (IDO1 and IDO2), and tryptophan-2,3-dioxygenase (TDO) utilize tryptophan as a substrate, and generate N-formylkynurennine [85]. N-formylkynurennine is rapidly converted to L-kynurenine (Kyn) by the action of formylaminase. Kyn can be further broken down into various biologically active metabolites, such as 3-hydroxykynurenine (3-HK), 3-hydroxyanthranilic acid (3-HAA), kynurenic acid (KA), xanthurenic acid, and quinolinic acid [85,86].

In a study exploring the relevance of tryptophan metabolism in coronary heart disease, free tryptophan concentration in the serum was found to be reduced and the Kyn/Tryptophan ratio was elevated in patients with coronary heart disease [87]. High concentrations of KA were also detected in unstable atherosclerotic plaques, suggesting the involvement of tryptophan metabolism in cardiovascular disease [88]. Detection by liquid chromatography-tandem mass spectrometry revealed that tryptophan catabolism was associated with ED, such that inhibiting IDO1 expression in ECs reduced the diastolic effect of vascular endothelium [89,90]. In insulin-resistant conditions, tryptophan depletion increases fatty acid oxidation and induces ED via the ROS pathway [91]. Interferon-γ also impairs glucose metabolism in the vascular endothelium by altering tryptophan metabolism [92]. Kyn, the first intermediate in the tryptophan degradation pathway, decreases EC viability in a dose-dependent manner [93]. 3-HK impairs EC viability by upregulating NADP-derived superoxide anions and accelerating EC apoptosis [94]. 3-HAA, a product with antioxidant effects produced during tryptophan metabolism, also exerts anti-inflammatory effects and protects ECs by inhibiting monocyte chemotactic protein-1 (MCP-1) secretion and VCAM-1 expression by promoting HO-1 expression [95].

Another important AA involved in the regulation of endothelial function is methionine. Methionine is an EAA obtained via the diet and is involved in the synthesis of proteins in the body. However, both an excess and deficiency of methionine affect normal vascular growth [52]. In a methionine loading experiment, a single intake of 100 mg/kg of methionine led to a significant increase in HCY levels in humans [96]. HCY causes oxidative stress by producing nitrotyrosine and decreasing endothelial NO concentration, leading to impaired vascular endothelial function [97]. Even a small intake of methionine (10 mg/kg) can induce ED [98].

L-phenylalanine promotes tetrahydrobiopterin synthesis and increases nitrite levels by activating guanosine triphosphate cyclase hydrolase (GTPCH, the first rate-limiting enzyme in tetrahydrobiopterin synthesis)/tetrahydrobiopterin (an essential cofactor for NOS metamorphosis, which helps stabilize the NOS dimer structure and increases the affinity of NOS for the substrate) pathway, thereby attenuating ROS and increasing NO levels, leading to improved endothelial function [99,100,101,102].

#### 2.2.2. Branched-Chain Amino Acids

BCAAs are important components of human skeletal muscle proteins and are mainly used as nitrogen carriers to assist the synthesis of other AAs required by muscles [103]. Recent studies have shown that BCAAs are clinically useful biomarkers for vascular diseases as their concentration is positively correlated with cardiovascular disease risk [104,105]. This may be related to the ability of BACCs to induce the expression of inflammatory factors and ROS through activation of the NF-κB pathway and oxidative stress, which inhibit the endothelium-dependent vasodilatory response and trigger ED [106]. In addition, the three BACCs—leucine, valine, and isoleucine—act independently. Leucine increases the synthesis of glucosamine, an inhibitor of endothelial NO synthesis, by enhancing rapamycin signaling and the expression of fructose-6-phosphate aminotransferase (the rate-limiting enzyme of glucosamine synthesis), thereby inhibiting NO synthesis in ECs and leading to vascular ED [107]. Interestingly, a related study showed that leucine supplementation alleviates atherosclerosis by suppressing inflammatory responses and regulating lipid levels [108,109]. Leucine also prevents hyperglycemia-induced ED by promoting insulin secretion [110], which is conflicting with previous findings. Leucine treatment has shown damaging effects in vitro and protective effects in vivo; this may be due to a more complicated leucine metabolic pathway in vivo or leucine treatment concentration. Further experiments are needed to gain clarity in this regard. Valine regulates lipid metabolism to protect the integrity of the vascular endothelium [108]. Isoleucine can interfere with angiogenesis by inhibiting VEGF production [111]. 

## 3. Amino Acid Homeostasis Disruption as a Risk Factor of Vascular Complications in Ischemic Heart Disease

AA homeostasis is crucial for the maintenance of normal physiological functions. Like other metabolites, AAs in the body are in relative equilibrium through AA intake, synthesis, and metabolism. Indeed, AA concentrations are much higher intracellular than in the plasma, suggesting that physiological concentrations of AAs in the cytoplasm and plasma can also be maintained through membrane transport. 

AA homeostasis is regulated by two signaling pathways, namely, the mechanistic target of rapamycin complex 1 (mTORC1) and GCN2/ATF4 pathways. A low concentration of AAs switches off mTORC1 signaling and upregulates ATF4 expression to induce the expression of AA synthesis genes to restore AA homeostasis [112,113]. When intracellular AAs are at normal levels, mTORC1 binds to GTPase Rheb to induce mTORC1 activation which in turn inhibits AA input [114,115]. The role of AAs as the main materials for peptide and protein synthesis is well studied and known, however, AA function as signaling molecules in the maintenance and regulation of metabolic homeostasis has received much attention in recent years.

### 3.1. Macrovascular and Microvascular Complications in Diabetes

Diabetes is an endocrine metabolic disorder characterized by elevated blood glucose due to impaired insulin secretion or utilization. Diabetes can result in a variety of complications, of which diabetic vasculopathy is the most common. The risk of vascular disease is 2–8 times higher in patients with diabetes than in normal subjects [116]. Diabetic vasculopathy includes macroangiopathy and microangiopathy. Diabetic macroangiopathy refers to diabetes-induced atherosclerosis of the aorta, coronary arteries, and peripheral arteries of the limbs, resulting in various cardiovascular diseases. Diabetic microangiopathy refers to diabetes-induced thickening of the microvascular basement membrane with hyaline-like material deposition, which leads to diabetic nephropathy, retinopathy, and diabetic neurological diseases. In clinical practice, the main cause of death in diabetic patients is diabetic vascular disease.

Patients with diabetes have abnormal glucose metabolism due to an absolute or relative lack of insulin in the body, which is also accompanied by abnormalities in lipid and protein metabolism, which mainly manifest as alterations in the circulating levels of many metabolites [117] (Table 1). Drabkova P et al. examined serum from patients with diabetes and found that levels of glycogenic AAs such as serine, glycine, arginine, threonine, and asparagine were significantly lower, while those of ketogenic AAs (including leucine and isoleucine) were higher than normal in diabetics [118], indicating that diabetes affects AA homeostasis. The higher levels of ketogenic AAs are likely due to elevated pyruvate dehydrogenase kinase isoform 4 (PDK4) expression in patients with diabetes that interferes with the conversion of pyruvate to acetyl coenzyme A, leading to the conversion of ketone bodies to ketogenic AAs [116]. Yuan et al. [119] applied Deuterium isobaric Amine Reactive Tag (DiART) labeling and liquid chromatography–mass spectrometry to detect the metabolic changes of intracellular amine derivatives in a diabetic macroangiopathy cell model and found that although there was no significant change in EC activity in ECs exposed to high sugar medium for a short period (0–6 h), the intracellular levels of alanine, proline, glycine, serine, and glutamine increased. When ECs were cultured in high glucose for 7 days, EC activity decreased and the levels of AAs metabolites increased. In another study, Disrupted glycine homeostasis ECs derived from diabetic human induced pluripotent stem cells (hiPSCs) showed disrupted glycine homeostasis [16]. 

The ratios of ornithine/arginine and ornithine/citrulline have been shown to be increased in the plasma of patients with type 2 diabetes and macroangiopathy [120]. Ornithine and citrulline are products of arginine metabolism and are formed under the action of arginase and NOS, respectively. Therefore, macroangiopathy is associated with arginine metabolism. Hyperglycemia induces increased arginase activity, which competes with NOS to bind arginine, leading to the decreased bioavailability of NO, ultimately leading to diabetic vascular dysfunction [120,124]. Furthermore, hyperglycemia raises ADMA levels, which compete with NOS to bind arginine, resulting in decreased NO production and aggravating diabetic vasculopathy [120,124].

Analyses of the Action in Diabetes and Vascular Disease: Preterax and Diamicron Modified Release Controlled Evaluation (ADVANCE) trial data revealed that the risk of diabetic macrovascular disease was positively associated with phenylalanine and negatively associated with histidine. Tyrosine and alanine were associated with the risk of diabetic microangiopathy events and negatively correlated with the risk of diabetic microangiopathy events [121]. Diabetes-induced renal impairment promotes tryptophan metabolism, leading to increased kynurenine production, leukocyte activation, oxidative stress, and inflammatory responses [122,123,125].

### 3.2. Hypertension

Hypertension is a leading cause of cardiovascular disease and death worldwide. Antihypertensive drugs and lifestyle interventions are currently being used to treat hypertension. Diet, especially the AA composition of dietary proteins, has been shown to play a key role in hypertension [123,126,127] (Table 2). In a study investigating the relationship between 24-h urinary BCAAs and blood pressure (BP) in elderly patients with hypertension, valine was found to be negatively correlated with systolic and diastolic BP and isoleucine was positively correlated with diastolic BP [128]. 

Mirmiran et al. [129] collected the dietary intake data of BCAAs from 4288 participants without hypertension and followed their clinical course for three years to determine the morbidity associated with hypertension and found that the intake of BCAAs, especially valine, was positively associated with the risk of developing hypertension. A high intake of aromatic AAs also led to a significantly higher risk of developing hypertension [130,131]. Methionine is metabolized to HCY, which impairs endothelial function inducing an increase in BP [132]. Under pathological conditions of hypertension, endothelial arginase activity is elevated and competes with NOS for binding substrate arginine, leading to decreased arginine concentration, uncoupling of eNOS, and impaired NO production, ultimately leading to endothelium-dependent diastolic dysfunction [83]. Recently, abnormalities in phenylalanine metabolism have been found in patients with primary hypertension, according to which inherited abnormalities in AA metabolism may be one of the key factors contributing to primary hypertension [136]. 

In another systematic analysis of eight observational studies, a negative association was found between plant protein intake and hypertension [137]. Tyrosine accelerates the release of norepinephrine and epinephrine in the central nervous system to exert hypotensive effects [133,134]. Tryptophan can regulate NOS activity through the synthesis of 5-hydroxytryptophan leading to a decrease in BP [133,138]; it can also be metabolized to kynurenine in ECs to activate adenosine and soluble guanylate cyclase to induce arterial diastole [139]. Glutamate inhibits oxidative stress and lowers BP by inducing glutathione synthesis [135,140]. Cysteine exerts its antihypertensive effects by regulating oxidative stress, glucose metabolism, insulin resistance, NO production, and glutathione synthesis [141]. Furthermore, low-protein diets during pregnancy have been shown to be consistent with increased susceptibility in offspring to hypertensive disorders [142]. Therefore, AA homeostasis disruption is closely related to the development of hypertensive disorders. 

### 3.3. Hypercholesterolemia

Hypercholesterolemia is mainly associated with a decreased clearance of low-density lipoprotein cholesterol (LDL-C) in the body. LDL-C binds low-density lipoprotein receptor (LDLR) and apolipoprotein B (ApoB) on the cell membrane to form a complex, which is endocytosed into the cell through the mediation of LDL receptor bridging protein 1 (LDLRA1). This complex is degraded within the lysosome and LDLR is recirculated to the cell membrane [143]. Mutations in genes encoding for the aforementioned proteins and enzymes lead to increased LDL-C levels in the body and hypercholesterolemia. 

Hypercholesterolemia is a direct cause of many cardiovascular diseases such as atherosclerosis, coronary artery disease, and stroke [144,145]. Currently, the clinical treatment of such cardiovascular diseases is often based on lipid-lowering drugs, such as statin but are ineffective in some patients with hypercholesterolemia [146,147]. Therefore, the search for novel treatments is urgent

In recent years, it has been found that hyperlipidemia is not only a disorder of lipid metabolism but is also associated with abnormalities of glucose metabolism and AA metabolism [148] (Table 3). Analysis of hepatic metabolic expression profiles of hypercholesterolemic rats revealed that a hypercholesterolemic diet leads to decreased levels of glycine, serine, threonine, and histidine and increased levels of asparagine and valine [149]. In addition, spermidine and S-adenosyl-methionine levels have been shown to be reduced in hypercholesterolemic rats, suggesting that hypercholesterolemia affects spermidine and methionine metabolism [149]. Similarly, another study revealed that the levels of spermidine decreased and those of spermidine metabolites such as ornithine and spermidine increased in hypercholesterolemia [150]. This is because the activities of high-affinity cationic amino acid transporters (CAT) and arginase on the surface of neutrophils are enhanced in hypercholesterolemia, where arginine enters the cell via CAT and is broken down into urea and ornithine by the action of arginase. Ornithine then enters the urea cycle as an intermediate or as a precursor for the synthesis of polyamines [5,83,151]. Hyperlipidemia-induced interferon-γ secretion by ECs activates IDO activity in macrophages and dendritic, which promotes tryptophan metabolism, leading to an increased kynurenine/tryptophan ratio and enhanced immune response, which may be an important mechanism of hypercholesterolemia-induced endothelial and renal impairment [5,150]. In addition, IDO and kynurenine are also involved in the immune response in the development of atherosclerotic disease by regulating IL-10 production [88].

## 4. Drugs That Restore Amino Acid Homeostasis for Improvement of Endothelial Dysfunction

Several drugs are known to restore AA homeostasis in ECs, including aspirin eugenol ester (AEE), folic acid, and *Astragali Radix* (Table 4). AEE, such as aspirin, is widely used in clinical practice because of its antipyretic and antiplatelet aggregation properties. Recently, AEE, formed by combining aspirin and eugenol, has been found to have stronger antithrombotic, antioxidant, and anti-atherosclerotic effects than aspirin alone. A study investigating the therapeutic utility of AEE in rats with acute blood stasis (ABS) [152] revealed that plasma concentrations of phenylalanine, isoleucine, valine, and tryptophan were significantly increased in rats in the ABS group. Consistent with this result, Zou et al. [153] also found an increase in many EAAs such as isoleucine, lysine, and valine in blood stasis rats. Phenylalanine can manifest vasoconstrictive effects through its conversion to tyrosine, a precursor of catecholamines, which can further exacerbate blood stasis [152]. This effect can be reversed by combined AEE treatment. In mice chronically fed a high-fat diet, body AA levels are significantly altered. In such mice, AEE can lower blood lipids and treat atherosclerosis by inhibiting the production of tyrosine metabolite hydroxyphenyllactic acid and tryptophan metabolite xanthurenic acid [154], increasing valine and leucine levels to promote the TAC cycle, reducing impaired energy metabolism [155], and normalizing abnormal AA metabolism to reduce oxidative stress and the inflammatory response [156] (Table 4). The protective effect of AEE on vascular endothelium can be achieved by the following: (i) inducing an increase in the level of methionine metabolic intermediate 5′-methylthioadenosine (A1 receptor agonist), which inhibits apoptosis of vascular ECs [157,158], (ii) increasing the level of indole acetaldehyde, a tryptophan metabolite, to inhibit EC proliferation [157,159], (iii) upregulating glutathione, L-lysine, and valine to regulate the oxidative stress response to promote vascular endothelial repair [157,160,161], and (iv) inhibition of iNOS expression and activity to reduce NO production [162].

HCY is a sulfur-containing AA produced during the metabolism of methionine; HCY is mainly metabolized in the body through methylation to methionine. Folic acid is an important cofactor in the metabolism of HCY. Folic acid deficiency results in the accumulation of HCY, which is excreted extracellularly causing elevated plasma HCY levels. High concentrations of HCY have been shown to be an important factor contributing to ED [166,167,168]. Folic acid supplementation improves endothelial function by promoting HCY metabolism, increasing NOS coupling, and improving NO utilization by EC [163,169,170] (Table 4).

*Astragali Radix* (AR) is the root of the legume *Astragalus mongolica*, which has been used as a drug for over 2000 years. According to the Pharmacopoeia of the People’s Republic of China, there are 163 Chinese medicinal preparations containing AR. *Astragalus membranaceus* tablets, extracts, and compound preparations are currently widely used in the treatment of cerebral infarction, coronary heart disease, and heart failure, as well as other cardiovascular diseases in clinical practice [171,172]. *Astragalus* saponin, the main component of AR, protects ECs from HCY damage by antioxidant stress and NO pathway modulation [164] (Table 4). Astragaloside IV downregulates inflammatory responses by inhibiting the NF-κB pathway, inhibits protein kinase C activation to improve EC barrier function, and activates the NO-cyclic guanosine monophosphate pathway to promote endothelium-dependent vasodilation [173,174,175]. The isoflavones in *Astragalus* inhibit apoptosis of ECs [176]. In addition, AR extract has been shown to promote EC migration, proliferation, and induction of angiogenesis through activation of the VEGF and PI3K/eNOS pathways [177]. AR has also been shown to ameliorate adriamycin-induced systemic multi-organ damage by inhibiting oxidative stress by modulating AA homeostasis [165,178,179,180]. Angiotensin converting enzyme2 and glucagon have also been shown to be involved in the regulation of AA homeostasis [181,182,183,184], but there is no evidence that they can improve ED by regulating AA homeostasis, and must be studied further.

## 5. Amino Acid Supplement for Improvement of Endothelial Dysfunction in Clinic

Due to its ability to produce NO in response to eNOS enzymes, arginine is thought to play an important role in the regulation of vascular endothelial function. The “arginine paradox” states that arginine intake should be proportional to NO blood concentrations. Although some clinical trials have shown that arginine supplementation improves endothelial function [185,186,187,188,189], other trials have failed to improve endothelial function [190,191,192,193,194]. We do not believe that the effect of arginine on improving endothelial function was related to the dose or duration of arginine supplementation because arginine was given in a wide range of doses (3 g/day to 20 g/day) and for a long period of time (1 day to 6 months) in these negative clinical trials [190,192,193]. In the previous section describing arginine, we mentioned ADMA, which competes with arginine for binding to NOS, resulting in a decrease in NO production. We suggested that arginine’s modulatory effect on endothelial function is related to ADMA levels in vivo. Monti LD, et al. treated 144 middle-aged patients with impaired glucose tolerance and metabolic syndrome for 18 months with L-arginine or placebo and found that L-arginine supplementation significantly reduced ADMA levels and improved endothelial function compared to the placebo group [195]. Several other clinical trials have shown similar results [196,197,198] (Table 5).

Taurine is a sulfur-containing AA that is widely distributed in all tissues and organs of the body [207]. Numerous experimental and clinical studies have shown that taurine has anti-inflammatory properties and that high levels of taurine can prevent cardiovascular disease [207,208]. In various cardiovascular diseases, such as hypertension, oral taurine (1.6 g/day) for 12 weeks significantly improved endothelium-dependent and non-endothelium-dependent vasodilation to lower BP, which may be related to taurine’s ability to inhibit calcium influx mediated by transient receptor potential channel subtype 3 in arteries [199]. In diabetic patients, continuous taurine supplementation (1.5 g/day) for 2 weeks improved ED [200]. Taurine has even been shown to improve vascular endothelial function in healthy people [201,202] (Table 5).

Tyrosine supplementation (150 mg/kg) was found to improve the contractile response of skin vessels to cold stimuli in a randomized, double-blind trial designed by James A Lang [203,209]. Citrulline supplementation influences vascular endothelial function via NO circulating metabolism to form arginine [204,205,206]. By promoting insulin secretion, leucine administration prevents hyperglycemia-mediated ED [110] (Table 5).

## 6. Current Challenge and Future Directions

The human genome encodes for 50 different AA [210,211], which is significantly higher than the number of protein-derived AAs. This suggests that there is competition and overlap between intracellular and extracellular AA function and transport. Although we know that the main biological role of AAs is incorporation into proteins, AAs and their metabolites are involved in several other processes, including energy supply, glucose, and lipid metabolism, as well as the regulation of hormone secretion under physiological and pathological conditions; however, the precise mechanisms underlying these roles are yet to be explored.

The metabolic pathways of multiple AAs are interconnected. It is necessary to clarify the potential cross-regulation between different AA-sensing pathways to fully understand the mechanism of action of each AA.

It is well known that the concentration of AAs in the cytosol is significantly higher than that in the blood [212,213]. This is achieved by antiporters and Na^+^-dependent symporters in the cytosolic membrane that maintains AA homeostasis [211]. However, AA roles and mechanisms in the development of ED and associated diseases, such as diabetes, hypertension, and atherosclerosis, are largely unknown. An integrated approach using genetics, metabolomics, and biochemistry may help characterize the mechanisms by which AAs regulate endothelial function.

Although primary human ECs are the cells of choice for studying ED, their use is limited by restricted sources and cell isolation techniques. ECs differentiated from disease-specific hiPSC lines reprogrammed from patient somatic cells are potential sources for disease modeling and drug screening and discovery in vitro [214]. We found that ECs differentiated from hiPSCs (hiPSC-ECs) derived from patients with T2DM (dia-hiPSC-ECs) have signature phenotypes of ED: disrupted glycine homeostasis, ED (increased protein expression of ICAM-1 and enhanced secretion of endothelin-1), increased cell senescence, and impaired mitochondrial function [16,215]. Thus, ECs differentiated from diabetic hiPSCs may be good cell models for studying and screening drugs for the treatment of ED.

## Figures and Tables

**Figure 1 cells-11-01372-f001:**
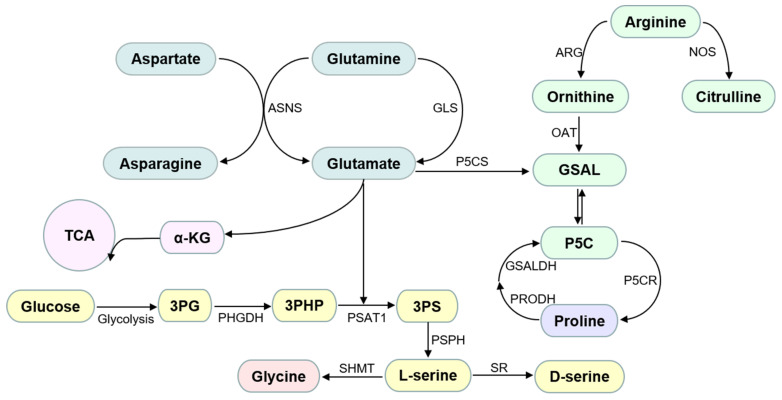
Schemetic diagram of AA metabolism in endothelial cells. Abbreviations used: α-KG: alpha-ketoglutarate; ASNS: asparagine synthetase; ARG: arginase; GLS: glutaminase; GSAL: L-glutamate-γ-semialdehyde; GSALDH: GSAL dehydrogenase; NOS: nitric oxide synthase; OAT: ornithine δ-amino acid transferase; P5C: 1-pyrroline-5-carboxylate; P5CS: P5C synthase; PHGDH: phosphoglycerate dehydrogenase; PSAT1: phosphoserine aminotransferase; P5CR: pyrroline-5-carboxylate reductase; PRODH: proline dehydrogenase; PSPH: phosphoserine phosphatase; SR: serine racemase; SHMT: hydroxymethyltransferase; TAC: tricarboxylic acid cycle; 3PG: 3-phosphoglycerate; 3PHP: 3-phosphohydroxypyruvate; 3PS: 3-phosphoserine.

**Table 1 cells-11-01372-t001:** Relationship between amino acids and diabetic vasculopathy.

Risk Factors	Experimental Model	Amino Acids	Findings	Reference
Diabetes	Patients with T2Ds	Serum AAs	Significantly decreased levels of arginine, asparagine, glycine, serine, threonine, and significantly increased levels of alanine, isoleucine, leucine, and valine in diabetics.	[118]
	Hyperglycemic human aortic ECs	AAs metabolism	ECs exposed to short-term hyperglycemia showed increased levels of alanine, proline, glycine, serine, and glutamine. AAs oxidative stress metabolites significantly increased when ECs exposed to glucose for 7 days.	[119]
	HiPSC lines from patients with T2Ds	Glycine	Dia-hiPSC-ECs had disrupted glycine homeostasis, increased senescence, and impaired mitochondrial function and angiogenic potential as compared with healthy hiPSC-ECs.	[16]
	Patients with T2Ds and healthy controls	Plasma AAs	The ratios of ornithine/citrulline and proline/citrulline were 60% and 95% higher, respectively, in patients with diabetes than in controls. The plasma ornithine/arginine ratio was 36% higher in patients with diabetes, indicating increased arginase activity.	[120]
	3587 men and women(a case-cohort study)	Plasma AAs (phenylalanine, isoleucine, glutamine, leucine, alanine, tyrosine, histidine, and valine)	Phenylalanine was positively associated with the risk of macrovascular disease, while histidine was inversely associated; higher tyrosine and alanine levels were associated with decreased risk of microvascular disease.	[121]
	Rats with experimental chronic renal failure	L-tryptophan levels and plasma concentrations in kidney, liver, lung, intestine, and spleen homogenates.	In animals with renal insufficiency, the plasma concentration and the content of l-tryptophan in homogenates of the kidney, liver, lung, intestine, and spleen were significantly decreased, while the plasma concentration and tissue levels of l-tryptophan metabolites in the kidney, liver, lung, intestine, spleen, and muscles were increased.	[122]
	859 patients with type 1 diabetes (baseline eGFR 30–75 mL/min/1.73 m^2^)	Plasma AAs	The patients showed decreased tryptophan/kynurenine, threonine, methionine, and tryptophan levels.	[123]

Legend: AA: Amino acid; EC: Endothelial cell; HiPSCs: Human induced pluripotent stem cells; Dia-hiPSCs: Diabetic human induced pluripotent stem cells; T2D: Type 2 diabetes.

**Table 2 cells-11-01372-t002:** Relationship between amino acids and hypertension.

Risk Factor	Experimental Model	Amino Acids	Findings	Reference
Hypertension	4288 participants aged 20–70 years without hypertension (3-year follow-up)	Dietary intakes of BCAAs (valine, leucine, and isoleucine)	Higher BCAA intake, particularly valine, is associated with a higher risk of incident hypertension.	[129]
	8589 Japanese subjects	Plasma AAs	Higher intake of aromatic AAs is associated with s significantly higher risk of developing hypertension.	[130]
	4287 adults (41.9% men), aged 20–70 years.	Dietary intake of AAs	High dietary intake of Leu.Ser/Thr.Trp ratio is associated with a higher risk of incident hypertension.	[131]
	172 South African adolescents (105 girls, ages 13 to <18 years)	Circulating HCY concentrations	Of these adolescents, 40% had elevated BP, of whom 37% fell in the lowest and 38% in the highest HCY tertiles.	[132]
	Normotensive or spontaneously hypertensive rats	L-Tyrosine, Tryptophan, Leucine, Isoleucine, Valine, Alanine, Arginine, and Aspartate	In spontaneously hypertensive rats, tyrosine (50 mg/kg) reduced BP by about 12 mmHg, while 200 mg/kg reduced BP by about 40 mmHg. Tryptophan injection (225 mg/kg) reduced BP in spontaneously hypertensive rats, but only by about half as much as an equivalent dose of tyrosine. Other AAs have no effect on BP.	[133]
	Spontaneously hypertensive rat	L-tyrosine	Intraventricular injection of 15 micrograms of l-tyrosine results in a significantly lower BP in the spontaneously hypertensive rat.	[134]
	4680 persons aged 40–59 years from China, Japan, the United Kingdom, and the United States	Dietary AA (glutamic, proline, phenylalanine, serine, and cystine)	Dietary glutamic acid (percentage of total protein intake) was inversely related to BP.	[135]

Legend: BCAA: Branched-chain amino acid; AA: Amino acids; Leu: Leucine; Ser: Serine; Thr: Threonine; Trp: Tryptophan; BP: Blood pressure.

**Table 3 cells-11-01372-t003:** Relationship between amino acids and hypercholesterolemia.

Risk Factors	Experimental Model	Amino Acids	Findings	Reference
Hypercholes terolemia	Hypercholesterolemic Wistar Rats	Liver AAs	A hypercholesterolemic diet resulted in decreased levels of glycine, serine, threonine, and histidine, and increased concentrations of asparagine and valine.	[149]
	Hypercholesterolemic Wistar Rats	Plasma AAs	A hypercholesterolemic diet led to a decrease in spermidine level and an increase in the level of the spermidine metabolites such as ornithine and spermidine.	[150]

Legend: AA: Amino acids.

**Table 4 cells-11-01372-t004:** Drugs that regulate amino acids homeostasis.

Medication	Experimental Model	Amino Acids	Findings	Reference
Aspirin eugenol ester	Blood stasis in rat	Plasma AAs	AEE treatment showed a favorable inhibition of the increase of phenylalanine, isoleucine, valine, and tryptophan.	[152]
	Hyperlipidemic rat	Plasma and urine AAs	AEE inhibits hyperlipidemia by inhibiting the production of tyrosine metabolite, hydroxyphenyllactic acid, and tryptophan metabolite, xanthurenic acid.	[154]
	Atherosclerotic hamster	Plasma and urine AAs	AEE promotes the TCA cycle and attenuates energy metabolism impairment by ameliorating blood lipid profile, reducing GLU and citric acid, as well as elevating the level of valine and leucine.	[155]
	Hyperlipidemia hamster	Liver and feces	AEE may improve lipid and bile metabolism, and reduce oxidative stress and inflammation, which were all beneficial for hyperlipidemia treatment.	[156]
Folic acid	126 patients with H-type hypertension	Serum HCY	After 3 months’ treatment with an FA dose adjusted according to methylene tetrahydrofolate reductase C677T genotype, HCY and ET-1/NO levels were significantly decreased in the intervention group and were lower than those after the first treatment phase and lower than in the control group (*p* < 0.01).	[163]
Astragali Radix	Acute phase endothelial dysfunction induced by HCY	HCY	AR and ASP protected endothelium-dependent relaxation against acute injury from HCY through NO regulatory pathways, in which antioxidation played a key role.	[164]
	Low-dose DOX-induced toxicity rat model	Rat brain AAs	The levels of six AAs, including glutamate, glycine, serine, alanine, citrulline, and ornithine, correlated with brain oxidative damage caused by DOX and rescued by AR.	[165]

Legend: AA: Amino acids; AEE: Aspirin eugenol ester; TCA: Tricarboxylic acid cycle; GLU: Glutamic acid; HCY: Homocysteine; FA: Folic acid; ET-1: Endothelin-1; NO: Nitric oxide; AR: Astragali Radix; ASP: Astragalus saponin; DOX: Doxorubicin.

**Table 5 cells-11-01372-t005:** Clinical study on the regulation of endothelial function by AA.

Amino Acid	Experimental Model	Dose	TreatmentTime	Findings	Reference
Arginine	Stable CAD patients	2 times/d (10 g/d)	4 weeks	Oral l-arginine supplement improved EF and reduced LDL oxidation in stable CAD patients.	[185]
	Healthy young smokers	3 times/d (21 g/d)	3 days	Oral l-arginine improves EF and vascular elastic properties of the arterial tree during the acute phase of smoking.	[186]
	Healthy male subjects	Intravenous l-arginine (10 g)	20 min	FMD assessment leads to impairment of EF by inducing an increase in ADMA, which is reversed by l-arginine administration.	[187]
	Healthy overweight adults with the HTW	3 times/d (4.5 g/d)	4 weeks	Supplementation with low-dose SR-arginine alleviates postprandial ED in healthy HTW adults when the baseline plasma arginine concentration is relatively low.	[188]
	Patients with peripheral arterial disease	50/100/500 mg l-arginine intra-arterially	once	Infusion of l-arginine increases blood flow and enhances the EF in diseased lower extremity human arteries.	[189]
	Patients with heart failure	20 g/day	28 days	Oral administration with l-arginine was ineffective in influencing EF in these patients with heart failure.	[190]
	Healthy males	20 g/day	28 days	Oral supplementation with l-arginine does not affect EF in normal healthy adults.	[191]
	Healthy young males	3 g	once	In healthy men, meal arginine only slightly enters the NO pathway and has no effect on basal EF.	[192]
	Patients with intermittent claudication due to PAD	3 g/day	6 months	In patients with intermittent claudication and PAD, oral l-arginine was less effective.	[193]
	Patients with severe malaria	12 g	once	L-arginine infused at 12 g over 8 h was safe but did not improve lactate clearance or endothelial NO bioavailability	[194]
	Patients with impaired glucose tolerance and metabolic syndrome	6.4 g/day	18 months	L-arginine increased the levels of EPCs and ADMA in subjects, suggesting that l-arginine can increase the expression levels of genes involved in metabolic and EF.	[195]
	Patients with CSX	0.125 g/min	120 min	Acute l-arginine infusion increases NO availability, decreases endothelin-1 levels, and improves EF in CSX patients.	[196]
	Clinically asymptomatic elderly subjects	3 g/day	3 weeks	Simvastatin does not enhance EF in subjects with elevated ADMA, but its combination with oral l-arginine improves EF in subjects with high ADMA.	[197]
	Patients with cardiovascular disease previously submitted to an aortocoronary bypass	6.4 g/day	6 months	Long-term oral l-arginine improves EF, decrease ADMA levels, and ameliorates insulin sensitivity and glucose tolerance.	[34,198]
Taurine	Prehypertensive individuals	1.6 g/day	12 weeks	Long-term taurine supplementation exerts antihypertensive effects by improving vascular function.	[199]
	Asymptomatic male diabetics	3 times/d (1.5 g/d)	2 weeks	Taurine supplementation reverses early, detectable conduit vessel abnormalities in young male diabetics.	[200]
	Healthy men	3 g/day	2 weeks	Taurine and Mg supplementation significantly increased EPC colony numbers and significantly decreased free radical levels in healthy men.	[201]
	Healthy men	6 g/day	2 weeks	2 weeks of taurine supplementation significantly increased vascular EF at rest.	[202]
Tyrosine	Young (25 ± 3 year) and older (72 ± 8 year)	150 mg/kg	once	Tyrosine supplementation was found to improve the contractile response of skin vessels to cold stimuli.	[203]
Citrulline	Healthy volunteers	2 times/d (0.75/1.5/3 g)	1 week	Oral l-citrulline supplementation raises plasma l-arginine concentration and augments NO-dependent signaling.	[204]
	Subjects with prehypertension	2 times/d (l-citrulline/l-arginine: 1.35 g/0.65 g)	6 weeks	WMJ supplementation improved aortic hemodynamics in middle-aged adults with prehypertension.	[205]
	Acute hyperglycemia in healthy adults	WMJ (500 mL/day)	2 weeks	WMJ supplementation improved FMD and microvascular function during acute hyperglycemia in healthy adults.	[206]
Leucine	Male volunteers	25 g	once	Leucine administration prevents hyperglycaemia-mediated ED probably due to enhanced insulin secretion.	[110]

Legend: AA: Amino acid; CAD: Coronary artery disease; EF: Endothelial function; LDL: Low-density lipoprotein; FMD: Flow-mediated dilation; ADMA: Asymmetric dimethylarginine; SR: Sustained-release; ED: Endothelial dysfunction; HTW: Hypertriglyceridemic waist; NO: Nitric oxide; PAD: Peripheral arterial disease; Mg: Magnesium; EPC: Endothelial progenitor cell; CSX: Cardiac syndrome X; WMJ: Watermelon juice.

## Data Availability

Not applicable.

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
