# Peer review of "The Role of Amino Acids in Endothelial Biology and Function"

_cells, 2022, doi:10.3390/cells11081372_

Round 1

Reviewer 1 Report

Manuscript ID: 1654450

Title: The role of amino acids in endothelial function

Summary

The purpose of the review paper is to summarize the role of amino acids in endothelial cells. Amino acids are important key regulators in endothelial cell function since they may regulate proliferation, migration, and survival. Therefore, they are importance in the vascular homeostasis and pathophysiology.

The topic is interesting to the field. Methods are appropriated and language is correct.

However, there are some points which need to be addressed or clarified.

Comments

The Authors reported some immune effect of amino acids. The mechanisms by which amino acid may interplay with immune system and the cellular pathway should be included.

The Author discuss the importance of amino acid for several diseases. Do amino acids may represent a valid biomarker of pathological condition or vascular damage?

There are some evidences that amino acids may play a role also in other disease such as obesity, cancer. and dementia. However, these topics are not discussed in the paper. I would suggest to include these important issues in the manuscript.

Tables are clear; however, figures representing cellular/molecular pathway involving amino acid would improve the quality of the manuscript.

Reviewer 2 Report

In the paper "The role of amino acids in endothelial dysfunction" the authors summarized the metabolic and signalling pathways of amino acids in endothelial cells and discusses the importance of amino acids homeostasis in the functioning of endothelial cells and vascular homeostasis, as well as, the possible directions for future research.

The paper could be of interest however some points should be addressed

  1. The paper needs language and grammatical revision in order to improve readability
  2. Add a paragraph on the clinical application of amino acids supplement in endothelial dysfunction 
  3. Add some figures for summarized some concepts expressed in the review 

Reviewer 3 Report

Thank you for allowing me to review your most interesting, very informative and excellently written manuscript on amino-acids (AA) and endothelial function (EF).

Paper includes well written overviews of the aminoacids and also takes the important step to decribing the pathophysological associations of the acids with the major clinical challenges: diabetes, hypertenion, dyslipideamia etc.

Moreover the steps from the accids and known and potential treatments is systematically provided. This makes the paper most relavant for clinicians.

The last paragraph of the paper on future directions again shares knowledge on the acids and at the same time contains some inspiring  philosophical notes; this is much appreciated.

Minor issues:

PDF page 1, line 31 'sate' = 'state'                                                                                                                                                                                                                                                                                                                                                                                                                                                                                                                                                                                                                                                                                                                                                                                                                                                                                                                                                                                                                                                                                                                                                                                                                                                                                                                                                                                                                                                                                                                                                                                                                                                                                                                                                                                                                                                                                                                                                                                                                                                                                                                                                                                                                                                                                                                                                                                                                                                                                                                                                                                                                                                                                                                                                                                                                                                                                                                                                                                                                                                                                                 a well written text on the physiology of EF and a detailed information section on the different AA's.

Some minor issues:

PDF page 1, line 31: 'sate' must read 'state'

Round 2

Reviewer 1 Report

Manuscript ID: 1654450

Title: The role of amino acids in endothelial biology and function

The purpose of the review paper is to summarize the role of amino acids in endothelial cells. Amino acids are important key regulators in endothelial cell function since they may regulate proliferation, migration, and survival. Therefore, they are importance in the vascular homeostasis and pathophysiology.

The topic is interesting to the field. Methods are appropriated and language is correct.

Figures and tables clearly represent the results.

In the revised version the authors have adequately and satisfactorily answered the issues previously raised.

Reviewer 2 Report

Thank to the authors for the correction of the manuscript. No further comments.